

# Tipping of the double-diffusive regime in the Southern Adriatic pit in 2017 in connection with record high salinity values

Felipe L.L. Amorim[1], Julien Le Meur[1], Achim Wirth[2], Vanessa Cardin[1]

[1]National Institute of Oceanography and Applied Geophysics - OGS, Sgonico, Trieste, 34010, Italy
[2] LEGI, Univ. Genoble Alpes, CNRS, Grenoble, F-38000, France

*Correspondence to*: Vanessa Cardin (vcardin@ogs.it)

**Abstract.** Whenever salinity decreases with depth the water column is either unstable or in a state called salt fingering (SF), which exhibits increased vertical mixing. Analysis of a high-frequency time series of thermohaline data measured at the EMSO-E2M3A Regional Facility in the Southern Adriatic Pit (SAP) from 2014 to 2019 reveal that in the south Adriatic SF is
the dominant regime. The same time series shows the presence of a very saline core of the Levantine Intermediate Water that penetrated with unprecedented strength during the winter of 2016/17 at around 550 dbar and even higher salinity water above. The effect of strong heat loss at the surface during that winter allowed deep convection to transport this high salinity water from the intermediate to the deep layers within the pit. This resulted in an increase in SF throughout the water column. In the subsurface layer (350 to 550 dbar) the increase is from 27% to 72% of observations. Consequently, we observe an alteration
of vertical stratification throughout the water column during winter 2016/17, from a stratified water column to an almost homogeneous water column down to 700 dbar, with no return in the following years.

## 1 Introduction

The Southern Adriatic Pit (SAP) is an important deep-water formation region. The formed dense waters in the area enter the
Eastern Mediterranean bringing oxygenated cold water to deeper layers. The stratification is altered by convection, gravity currents, lateral intrusions from neighboring basins and (double diffusive) mixing. The first two are intermittent strong events, while the last one is continuous with varying magnitude. During winter, vertical convection can occur, destroying density barriers throughout the water column. These combined processes allow efficient mixing and exchange of properties between the upper, intermediate and sometimes also the deep layers, resulting in changes in the properties and stability of water
characteristics (Cardin and Gačić, 2003; Leaman, 1994; Leaman and Schott, 1991; Mertens and Schott, 1998; Vilibić and Orlić, 2002, Cardin et al., 2011). Intermittent gravity currents bring cold water masses from the northern Adriatic to the deep



parts of the SAP, a process that is affected by topographic canyons and which restores the bottom to surface density difference (Chiggiato et al., 2016).

The water mass characteristics in the Southern Adriatic leads to a double-diffusive salt finger process (SF, the temperature
stratification is stable, while the haline stratification is unstable) because of the warm and salty Levantine Intermediate Water (LIW) overlying the colder and fresher deep Adriatic water (AdDW). Indeed, double-diffusive convection occurs in the ocean when either the temperature or salinity-induced stratification is statically unstable, while the overall density stratification is statically stable (Stern, 1960). The potential energy bound in the unstable component is released by molecular diffusion, which is 100 times faster for heat than for salt (Schmitt, 1994; Stern, 1960). McDougall et al. (1988) describe the vertical stratification
in terms of the following stability regimes: SF, diffusive convective (the temperature stratification is unstable, while the haline stratification is stable), doubly stable and statically unstable. According to You (2002), large areas of the world's oceans are favorable to double diffusion, including the Mediterranean and the Adriatic Seas. Specifically, SFs have been observed in the Mediterranean Sea (Schmitt, 1994; Meccia et al., 2016, Menna et al., 2021), the Tyrrhenian Sea (Tait and Howe, 1968; Durante et al., 2019) and the Adriatic Sea (Carniel et al., 2008). Double-diffusion studies in the oceans typically use CTD profile
transects acquired at high frequency (hours) but over a short period (days to weeks). They also provide a fine vertical resolution but are not able to depict a continuous evolution of double diffusion over several years. Durante et al. (2019) and Menna et al. (2021) both used a longer time series of Argo data profiles in the Tyrrhenian and Ionian/Levantine Seas, respectively, their temporal resolution of the analyzed data ranged from weeks to months. In this work, we examine high-frequency, long-time data from the EMSO-E2M3A Regional Facility (http://emso.eu/observatories-node/south-adriatic-sea/) that allows us to assess
the temporal evolution of double diffusion over long-time scales at high frequency (hours), but only at a coarse vertical resolution at a fixed horizontal location. Fine scale double-diffusive staircases typically extending O(10 m) vertically cannot be directly observed with our data, but vertical temperature and salt stratification leading to double diffusion can be observed.

In the SAP, the saltier Levantine Intermediate Water (LIW) forms intrusive features (Gačić et al., 2010; Vilibić and Orlić, 2001; Vilibić and Supić, 2005). Double-diffusive mixing processes acting at the interfaces of these intrusions alter heat and
salt transport, but turbulence levels must be sufficiently low for double diffusion to act (Lee et al., 2014). Cardin et al. (2020) estimated a high vertical bulk diffusivity coefficient of 5x10-4 m2/s based on two types of data i.e 13-year time series of observational data (2006-2019) of temperature from the EMSO E2M3A observatory and available vertical profiles (1985-2019) in the area. The differences in the molecular diffusivity of temperature and salinity can enhance vertical mixing through double diffusion, resulting in effective diffusivities on the order of 10-4 m2/s (Radko, 2013). Even though this process happens
on molecular scales, it can affect larger spatial extents by mixing water mass properties. Double diffusion is particularly wide-spread in the main thermocline (Radko, 2013) and contributes to the vertical transport of heat, affecting the global climate due to the air-sea fluxes occurring in the sea surface. The prevalence of double diffusion in the upper ocean can enhance the mixing of nutrients, which directly controls biological productivity (Radko, 2013; Meccia, et al., 2016). Despite their potentially





important contribution to vertical mixing, there is no quantitative work that has investigated the evolution of double diffusion
regimes using continuous long-term datasets (multi-year) with a high frequency (hours to days). In this study, we describe the
changes of double-diffusion regimes using time series data obtained from the EMSO-E2M3A in the SAP. Description of the
experimental data and methodology applied is described in the following section while the analysis and discussion are
presented in section 3. Conclusions are given in section 4.

## 2 Data and methods

The data used in this study were obtained from the EMSO South Adriatic deep observatory that is one of the regional
infrastructures established in 'key environmental sites' in European seas within the framework of the EMSO ERIC network.
The observatory consists of two sites, the first located in the center of the South Adriatic Pit (E2M3A, Fig. 1) and the second
on the west side of the escarpment (BB and FF). The data collected at the E2M3A site enables the monitoring of changes that
can be linked to variations in the sub-mesoscale, mesoscale and general circulation of the Eastern Mediterranean Sea or, on a
larger time scale, to the climate variability in the area, demonstrating the importance of high-frequency measurements to
resolving events and rapid processes (Cardin et al., 2020) as well as their long term occurrence and modulation.

In recent years, the observatory has allowed monitoring of convective processes and the formation and arrival of dense water
(crucial for oxygenation in the deep sea). The EMSO E2M3A site (position 18.06°E, 41.53°N) has operated almost
continuously since 2006 (Bensi et al., 2014; Cardin et al., 2015; Cardin et al., 2018, Cardin et al., 2020a;
http://emso.eu/observatories-node/south-adriatic-sea/). The EMSO-E2M3A dataset used in this study consists of potential
temperature (θ) and salinity (S) data collected hourly at seven vertical levels (150, 350, 550, 750, 900, 1000 and 1200 dbar)
ranging from November 2014 to October 2019; this was chosen considering the longest period available between two
maintenance cruises with calibration of instruments.

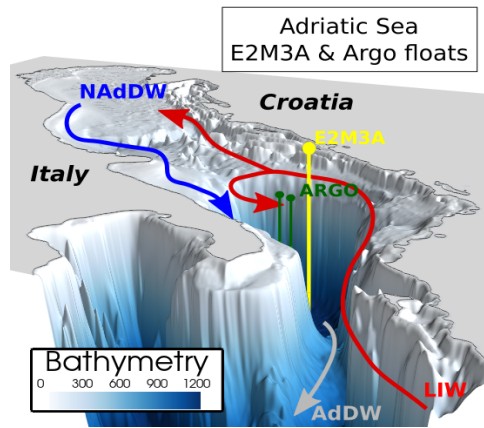


**Figure 1: E2M3A mooring location in the South Adriatic pit (SAP; yellow dot) and Argo float positions (green dots), with the main
water masses routes.**



Derived parameters such as θ, Turner angle and squared buoyancy frequency (hereafter referred as Tu and N² respectively)

were calculated using the Gibbs Seawater (GSW) Oceanographic Toolbox containing the TEOS-10 routines (McDougall and Barker, 2011). Calculations were achieved using the Numpy module (Harris et al., 2020), the Pandas module (McKinney, 2010) and the Scipy module (Virtanen et al., 2020) from Python. The different figures presented here were obtained thanks to the Matplotlib module (Hunter, 2007) and to the Mayavi module (Ramachandran and Varoquaux, 2011) again from Python. The squared buoyancy frequency (N²) values are calculated considering 33h-filtered conservative temperature (CT) and 33h-

filtered absolute salinity (SA) (McDougall and Barker, 2014) and are centered at the mean pressure between two of the mooring time series, i.e. at 450, 650, 825, 950, 1100 dbars (we removed the 150 dbar layer as important gaps in this time series are present). Interpolation is then performed to obtain a visualization of N² on the whole water column. In order to explore if the water column was undergoing double-diffusive convection and its related local stability we estimated the Turner angle (Tu) defined following Eq.(1):

$Tu=\tan^{-1}(\frac{\alpha\partial\theta}{\partial z} - \frac{\beta\partial S}{\partial z}), (\frac{\alpha\partial\theta}{\partial z} + \frac{\beta\partial S}{\partial z})$ see (Ruddick, 1983),

where tan-1 is the arctangent of the fourth quadrant; α is the coefficient of thermal expansion; β is the equivalent coefficient for the addition of salinity, sometimes called the "coefficient of salt contraction"; θ is the potential temperature; and S is the salinity. The Turner angle, Tu, compares the density stratification due to the temperature gradient with that due to the salinity gradient (van der Boog et al., 2021). The types of instability are determined by the signs of the potential density and salinity

gradients rather than by their absolute values (Meccia et al., 2016).

When Tu is between -45° and -90°, diffusive-convective double diffusion is possible; for Tu between -45° and 45°, the water column is doubly stable, meaning that the water column is stably stratified with respect to both temperature and salinity; and for Tu between 45° and 90°, SF double diffusion is expected (You, 2002). To calculate Tu only data periods that covered all available vertical levels were used to achieve the best discretization of the vertical gradient by using the values of two

consecutive levels. Therefore, we considered for this purpose the time series at the 7 levels ranging from November 2014 to October 2019. To evaluate further the water mass-properties in the vertical, we calculated the vector length (VL) defined following Eq. (2):

$$VL = \sqrt{(\alpha\frac{dT}{dz})^2 + (\beta\frac{dS}{dz})^2}.$$

To the best of our knowledge the VL has not been discussed in connection with stratification and the Tu. A higher VL indicates

an increased change of water-mass properties and therefore emphasizes the importance of Tu. On the other hand, when the VL is small the water column is essentially unstratified and changes in the Tu insignificant.

Complementing the time series data of T and S, we used the two Argo float number 6903197 profiles (https://www.euro-argo.eu/Argo-Data-access), daily ERA5 net surface heat fluxes (Qtot; Hersbach et al., 2020) from the European Centre for





Medium-Range Weather Forecasts (ECMWF) and mixed layer depth (MLD), provided by the Copernicus Marine Service at

the nearest grid point to EMSO-E2M3A. The winter period is considered to run from December to February. Heat flux

variations were determined by averaging Qtot over an area containing points 0.3° from E2M3A; this yields an average of 4

points. The MLD was calculated using its maximum in an area closer than 0.1° from the mooring location.

Surface relative vorticity defined following Eq. (3):

$$RV = \frac{\partial v}{\partial x} - \frac{\partial u}{\partial u}),$$

was obtained using geostrophic velocities within the isobath of 1150 dbar, resulting in a 18 points average. It is important to

note that the EMSO-E2M3A deep mooring is very close to the center of the SAP, where the relative vorticity has a smaller

variance than at the edges on the 1000 m isobath.

The two Argo float (# 6903197) profiles used here are profile 81 (Jan. 16, 2017) and profile 82 (Jan. 21, 2017), which are

located about 38 and 46 km (respective positions are (41.52°N;17.64°E) and (41.40°N;17.57°E)) of the mooring (Fig. 1). The

Argo profiles of potential temperature and salinity were first linearly interpolated to 1 dbar vertical resolution and the Turner

angle was calculated using the 20-point moving-averaged profiles to avoid excessive noise in the results.

## 3 Analysis and discussion

### 3.1 Thermohaline variability in the area

The Hovmöller diagrams, based on time series of potential temperature and salinity measured at SAP (Fig. 2), show a clear

change in the characteristics before and after 2017. Looking at the 5-year data set used for this work (November 2014-October

2019), the water column from the end of 2014 to the end of 2016 shows a clear signature of LIW occurrence restricted to the

layer between 400m and 600 m, with a core value of 38.84. Unfortunately, the lack of data from the 150 dbar time series

prevents defining the characteristics of the upper layer between November 2014 and November 2015, although there is

indication of an intrusion of less saline water into the LIW horizon which seems to be the cause of the deepening of the LIW

core to around 550m by mid 2015, also reducing its thickness. In October 2016, the surface layer was extraordinarily filled to

about 300 dbar by very salty and warm waters, presumably an intrusion of LSW (Levantine Surface Water), with values above

38.90 (since no data were available above 150 m, the value could not be accurately determined). These two cores with high

salinity were also observed by Mihanović et al. (2021) based on Argo float data in the same area. By December of 2016, the

deep salinity maxima mixed with surrounding waters and deepened to about 800 dbar with a reduced value of around 38.82.



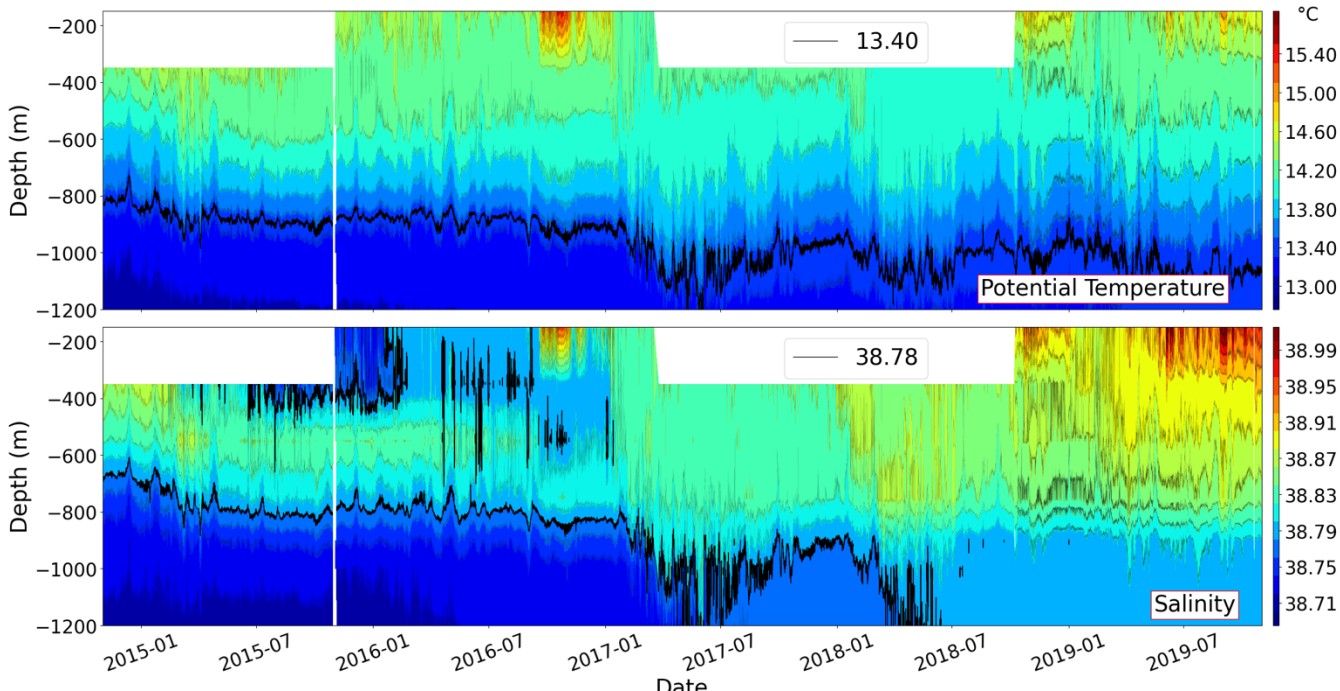

**Figure 2: Hovmöller diagrams for θ (°C; top) and S (bottom) for the studied period of the EMSO-E2M3A mooring data. The isotherm of 13.4 °C and the isohaline of 38.78 (black solid line) show the mentioned strong oscillations after the winter 2016/17.**

In early 2017, the salinity at the surface and in the intermediate layer changed radically. The most important feature is the intrusion of high salinity into the intermediate layers and a strong convection event in early winter, which is confirmed by the pool of homogenized water to a depth of about 700 dbar in the pit (Fig. 3b). Indeed, strong heat losses occurred in January (700 W/m$^2$) and in March (500 W/m$^2$) (Fig. 3a), which, together with the contribution of salt in the water column, facilitated the erosion of the stratification (Fig. 3b, Table 1). Mihanović et al. (2021) reported very high salinity values (above 38.9) in

the upper 100 dbar in the Southern Adriatic as early as mid-March from Argo data, which increased to an exceptional salinity and temperature maximum at and near the surface by summer. Throughout the year, salinity in the water column remained exceptionally high, confirming the same trend observed by Mihanović et al. (2021) along the Palagruža transect during summer and autumn. Moreover, temperature and salinity observations are higher than the values/data reported in previous climatologies for the Adriatic and in the literature (Buljan and Zore-Armanda, 1976; Artegiani et al., 1997; Lipizer et al., 2014). In 2017 the

water column showed a two-layer structure divided by the isohaline value of 38.83, with high salinity waters occupying the surface and intermediate layers (down to about 800-850 dbar) and less saline waters (about 38.78) filling the deeper part of the pit. Water temperature in the latter, usually lower than 13°C, increased to ≈13.2°C. The strong oscillations (Fig. 2) occurring during this period below 750 dbar are not directly related to salt fingering and will be discussed in an upcoming publication (Le Meur et al. in preparation).




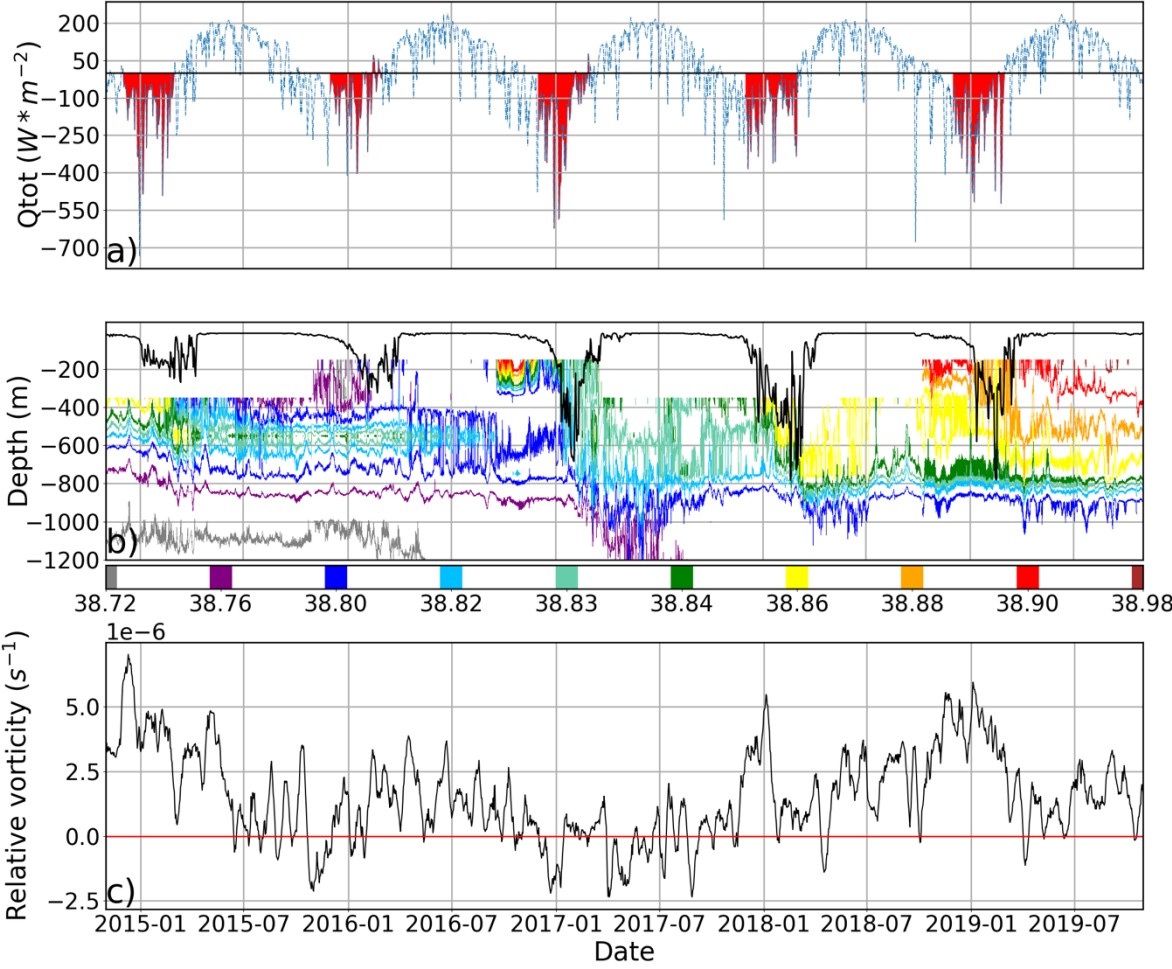

**Figure 3: The evolution of the three forcings connected to the changes in buoyancy. a) Qtot (W*m-2), b) Salinity with mixed layer depth (MLD (m); solid black line) and c) RV (s-1).**

Surface salinity in the Southern Adriatic peaked between August and October 2017, as indicated by data from two floats in the Southern Adriatic (not shown). In early 2018, another sharp increase in salinity and temperature was observed in the surface and intermediate layers, reaching values of 38.89 and 14.25°C, respectively, predisposing the water column to convection. Integrated surface heat loss for the winter period (Table 1) showed that despite less severe heat losses in the winter of 2018 (but very frequent weak losses in February and March), the new contribution of salt triggered again the convective mixing of

the water column to about 850 dbar (Fig. 3b).

.



| Winter | Integrated fluxes [J/s] | Maximum heat loss [W/m²] | Mixed Layer depth [dbar] |
|---|---|---|---|
| 2014 - 2015 | -1.14e+09 | -736 | 270 |
| 2015 - 2016 | -6.75e+08 | -411 | 325 |
| 2016 - 2017 | -1.22e+09 | -623 | 684 |
| 2017 - 2018 | -1.01e+09 | -385 | 782 |
| 2018 - 2019 | -1.35e+09 | -523 | 781 |

**Table 1: Integrated heat fluxes for winter period (December - February, area in red, Fig.3a), maximum heat loss and the depth reached by the mixed layer during the convection.**


The water column maintained its "two-layer structure" in terms of salinity, with a halocline with values between 38.82 and 38.80 separating the surface layer from the deep layer. The deep layer also experienced a sharp increase in salinity with values between 38.76 and 38.78, which had never been observed in the last decade. The oscillations observed in the deepest layers in the previous year were present also after March 2018. Inflow of high saline and warm waters in the pit was registered closer

to the surface in summer, which slowly deepened its horizon in early 2019 and completely filled the surface and intermediate layers with values around 38.88 during the winter period (end of February - beginning of March 2019). There is no convection observed in the winter 2019. This might be a result of SF, which increases salinity in the intermediate layers, thereby avoiding an unstable stratification. No substantial changes were observed in the deep layer which maintained the same characteristics as in late 2018 throughout 2019 and until the end of the study period. The above discussion indicates that saltier waters

overlying less saltier waters are a recurrent feature over the whole water column in the SAP. This leads to SF or convection.

### 3.2 Abrupt squared buoyancy frequency changes

The Hovmöller plot of $N^2$ shown in Fig. 4 shows a change in the stability structure of the water column in the SAP before and after the beginning of 2017. In winter 2015, the water column had high stability at about 450 dbar and 800 dbar, which

prevented convection despite high integrated heat losses. The barrier of high stability at about 450 dbar was eroded in 2016 due to saline intrusions in the upper layer. During winter 2016/17, the water column stability weakened further due to strong surface cooling, also pushing the core of the stable middle layer, which was around 800 dbar, down to about 950 dbar. After the winter of 2016/17, the $N^2$ structure did not return to the state of previous years. Three forcings favored the change of the



N² structure in 2017, as shown in Fig. 3a-c: First, we observed a strong heat loss at the surface during the winter of 2016/17,

with the high salinity below the surface (150 to 350 dbar) pushing the intrusion of the LIW core downward and the departure of the less saline water layers above the LIW (second). A change in relative vorticity (third) that shifted from cyclonic (1.7e-6 s⁻¹), favoring preconditioning, to anticyclonic (-2.3e-6 s⁻¹), possibly leading to upward Ekman pumping that brought more saline water to the surface, followed by a sharp jump back to cyclonic due to sinking of the dense waters. These three factors contributed to the observed SF and low N².


**Figure 4: Hovmöller diagram of squared buoyancy frequency (N²; s-2). The layers with data (at mid-pressure between two observation depths) are marked with thin solid lines (450, 650, 825, 950 and 1100 dbar). A clear change in N² after the winter 2016/17 is noticed.**


### 3.3 Double diffusion and its evolution

Ocean stratification is subject to abrupt changes due to intermittent events such as gravity currents, horizontal intrusions and convection, as discussed in the previous subsection. These events not only overlap the continuous double-diffusive mixing, but they also interact with each other. In this section, we emphasize the effect of changes due to the convective process on the

double-diffusive regime. We characterize the double diffusion by the Turner angle, Tu, and vector length, VL, introduced in



section 2. During the 5-year period, the Tu time series (Fig. 5a-e) showed variability ranging mainly from SF to doubly stable regimes, with the SF regime (blue line in Fig. 5a-e) dominating below a depth of 550 dbar.

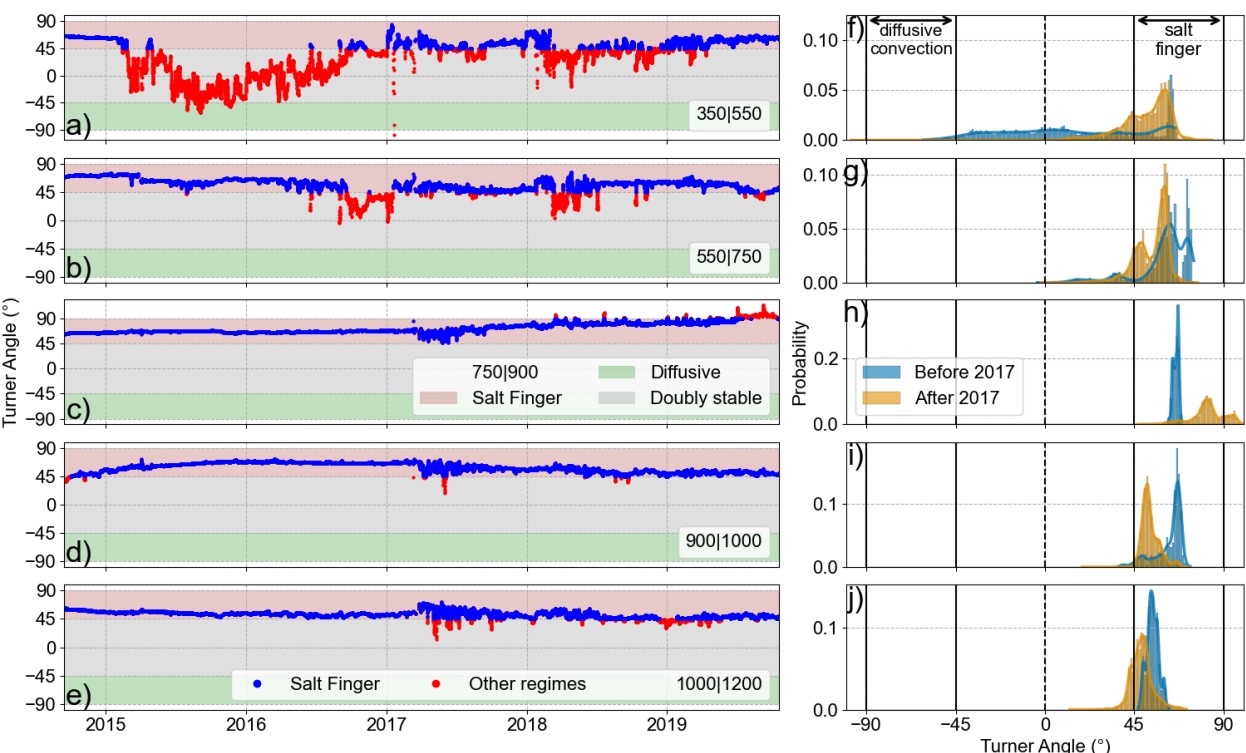

**Figure 5: a-e) Turner angle time series using vertical potential temperature and salinity gradients and f-j) the related PDFs calculated before (blue) and after (orange) 2017. The numbers in the legend represent the pressure levels that define the layer and blue markers are periods of SF regime (pink background).**

The uppermost layer is SF at the end of 2014 due to low salinity at 550 dbar. With the decrease of salinity at 350 dbar and an increase at 550 dbar (see time series in appendix), the uppermost layer becomes doubly stable from late winter 2015 throughout 2016. At the same time, the potential temperature at 550 dbar increases and the stratification in both variables decreases, as can also be seen in VL in Fig. 6a. The stable stratification leads to low values of vertical mixing and the upper layer, and the layers below evolve more independently. Due to an increase in salinity at 350 dbar in early 2017, the uppermost layer returns to strong SF and convection occurs, decreasing VL. The upper layer is homogenizing. This scenario of increased salinity in the upper part of the layer, leading to SF, convection and consequently a decrease in VL, is also occurring with a time lag in the adjacent lower layer and is repeated in 2018. A further increase in salinity in 2018 and 2019 leads to a strong SF behavior at low VL values, which is only sporadically interrupted by an increase in temperature at 350 dbar.



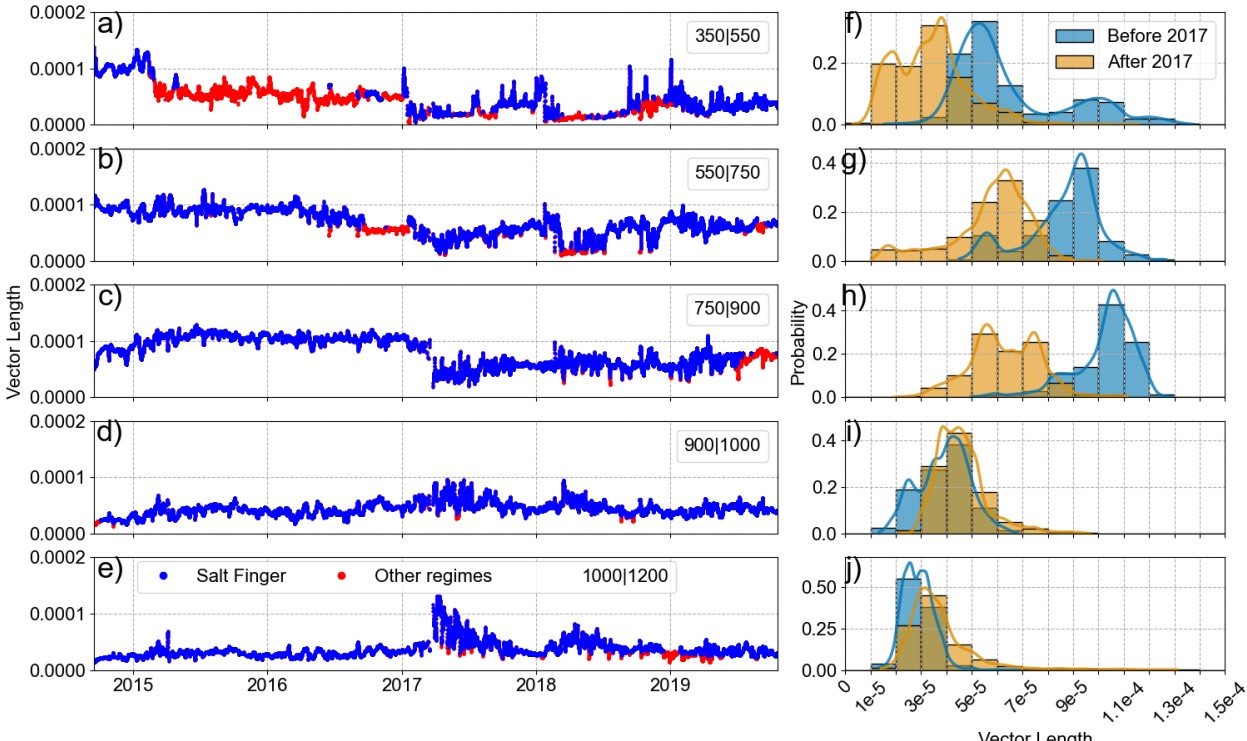

**Fig. 6: a-e) Vector length (1/m) time series and f-j) the related PDFs.**


The second layer (550-750 dbar, Figs. 5b and 6b) is in a SF regime, interrupted only in late 2016 and spring 2018. In the first period, a reduction in salinity at 550 dbar leads to stabilization. This reduction is not observed in the adjacent layers and results from the deepening of the LIW layer, which also reduces the stability of the upper layer. In the second period, the stabilization is due to a sudden increase in salinity at 750 dbar, which is observed to varying magnitude throughout the water column. Since

it strongly reduces the VL down to 900 dbar, it is due to convective mixing.

The third layer (750-900 dbar, Figs. 5c and 6c) shows a continuous increase of Tu from SF to unstable by the end of 2019 due to a continuous increase of salinity at 750 dbar. There is a sudden decrease of VL due to the 2017 event, caused by an increase in salinity at 900 dbar.

In the two deepest layers (900-1000 and 1000-1200 dbar, Figs. 5d, e and 6d, e), there is a shift towards reduced SF strength

after the 2017 event, which is also seen by increased variability of VL. Analysis of VL shows an eradication of stratification in early 2017 from 350 dbar down to 900 dbar which persists until the end of the data record. The prominent peak in Tu and VL in early 2017 is due to the arrival of hot, salty water at 350 dbar that triggered strong SF (see also Fig. 7 and the discussion of the Argo data below).  The stabilization of the water column in early spring of 2017 in the two lower layers, seen by a jump and oscillations in the VL are due to gravity currents (Le Meur et al. in preparation)



We also plotted the probability density functions (PDF) of Tu and VL through the end of 2016 and from 2017 through the end of the observation period (Figs. 5 f-j and 6 f-j). All PDFs from Tu show dominance of the SF regime except in the upper most layer due to the period from early 2015 to the end of 2016. The regime change in 2017 is indicated by an increase in Tu in the 750-900 dbar layer and a decrease in the two lowest layers; in all three layers, the peaks of the pdfs are well separated, indicating a regime change. The Tu shows significant destabilization of the 750-900 dbar layer after 2017 and a stabilization

of the layer below, due to the convection event in 2017.

The PDFs of VL show a significant decrease above 900 dbar, indicating the eradication of stratification; the PDFs are well separated for the two layers between 550 dbar and 900 dbar.

The SF occurrence was 83% of all measurements. The most relevant change in SF occurrence is for the upper-most layer (350|550 dbar), changing from 27% to 72% for measurements before and after 2017 respectively, indicating the significant

increase of SFs (Fig. 5a, blue dots).

Our observational data shows a shift in the vertical double-diffusion regimes in the SAP during the winter 2016/17 in all layers (Fig. 5), in the Tu and VL variables. During this period our data from EMSO-E2M3A show SF from 350 dbar to the bottom (Fig. 8K and l) at a VL which is significantly non-vanishing (Fig 6a-e). These results are confirmed by the Argo float section (Fig. 7 g and h) (van der Boog et al., 2022). The Argo floats have a fine vertical resolution and clearly show the existence of

staircases (Fig. 7 e-f) (van der Boog et al., 2022). The stair-case structure of SF has thin interfaces separating thicker, well mixed, layers. Salt flux from the upper high salinity layer to the low salinity layer below continuously mixes the water, and the convective turbulence of the layers limits the length of the fingers (Schmitt, 1988; Schmitt, 2003; Carniel et al., 2008; Durante et al., 2019).




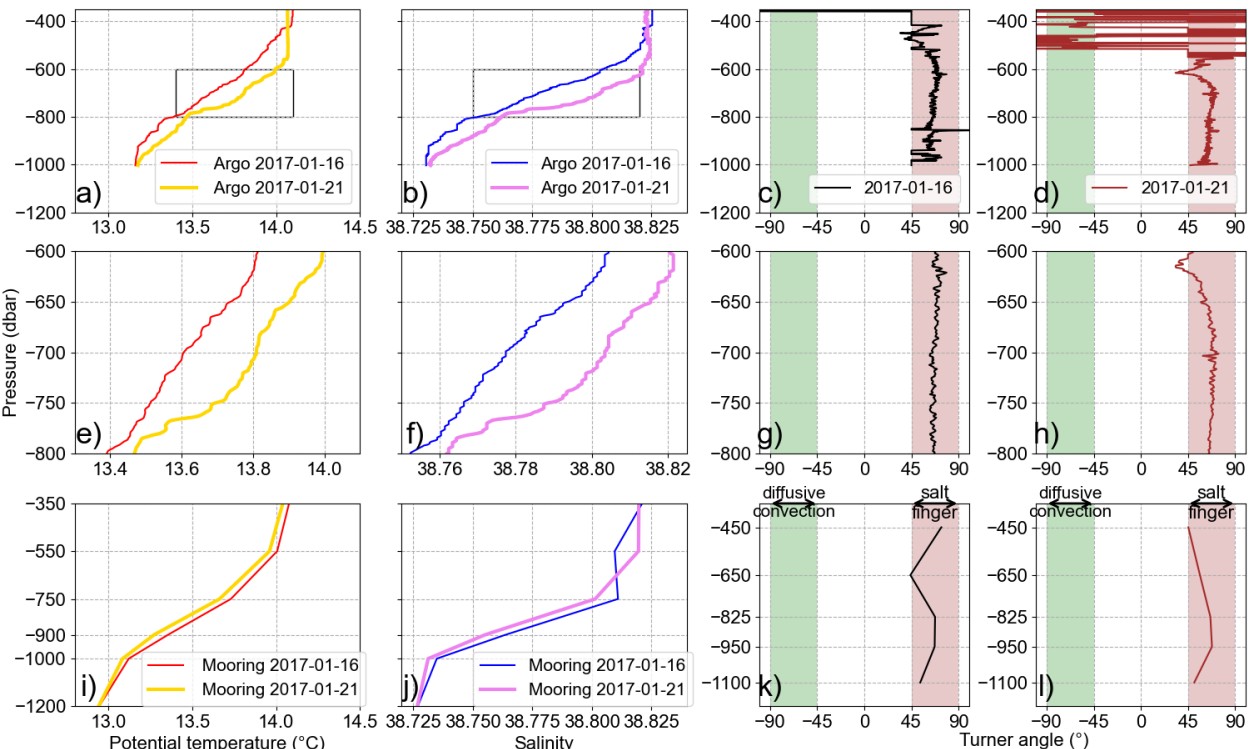

**Figure 7: a-b) Argo float θ and S profiles on January 16 and 21, 2017, with rectangles defining the area to be zoomed. (c-d) Turner angle calculated for Argo profiles. (e-h) Zoom of Argo profiles and Tu between 600 and 800 dbar showing staircase features below convection depth in the SF regime. (j-l) Mooring data collected on 16 and 21 January 2017 showing good coherence with Argo data.**

The increase in near surface salinity by horizontal advection through the strait of Otranto (Kokkini et al., 2020; Mihanović et al., 2021) favors SF, followed by convection down to 500 dbar on Jan 21st, 2017. We go further and hypothesize that the high salinity at the surface of the SAP could influence the observed change in relative vorticity through a salt lens of warm waters, explaining the anti-cyclonic tendency (see i.e., Tokos and Rossby, 1991, for subsurface salt-lens dynamics) during the preconditioning phase in the winter of 2016/17. The strong wintertime cooling then leads to convection and the sinking of the salt lens, which increases the relative vorticity. Below the convection depth the SF strength persists. In contrast with the results of Meccia et al. (2016), we observed an increase of SF development in the SAP due to the increase in salinity, enhancing salt-finger double diffusion in the subsurface layer of 350-550 dbar. The results of Carniel et al. (2008) showed that the northern part of the Adriatic works differently than the SAP because of the Po River plume influence, which brings fresh water to the surface and allows the presence of diffusive-convective double diffusion, not significantly observed in the SAP in the analyzed data. Still, the same order of magnitude for diffusivity coefficient was estimated (O(10-4 m2/s)).





## 4 Conclusions and perspectives

SF is the dominant regime at the EMSO-E2M3A facility. From 2014 to the end of 2016 strong stratification above 400 dbar and at 800 dbar reduced the mixing between the upper ocean and the deep SAP. After the winter of 2014/15 the upper layer passes from SF to doubly stably stratified further reducing the vertical mixing. A shift in the SF characteristics is observed in the winter 2016/17 throughout the water column due to the arrival of high saline waters leading to a convective event penetrating to about 750 dbar. This extreme event increases the SF above 900 dbar and homogenizes the water column above.

No return to the previous state is seen in the observation period. This might possibly happen on a decadal time scale, as it was determined in Cardin et al. (2020) that the characteristic time-scale of the deep SAP is around 7 years. Continued observations are necessary to conclude on a return to the previous state. Salt fingering is shown to be a consequence and a driver of the density structure in the SAP: it is governed by salt intrusions through the strait of Otranto and increases the vertical mixing. Salt intrusions from the Ionian favor SF and convection and homogenize the water column in the SAP. Dense gravity currents

originating from the northern Adriatic have the potential to enhance the stratification by increasing the density in the SAP. The vertical exchange is performed by mixing enhanced by SF and convection. How the competition between sporadic-extreme and continuous processes changes the structure of the water column and, in the future, how it acts on the thermohaline circulation of the Mediterranean is not decided.

To the best of our knowledge, it is the first time that a high-frequency multi-year time series from different water depths is

used to determine the Turner angle and the double-diffusive regimes. The supplementary analysis of vector length to verify the significance of Turner angle results was a novelty introduced in this study, and analyzed together with the squared buoyancy frequency allowed us to discuss the stability structure changes. The increase in SF observations after the winter 2016/17 cannot be ignored because it can modify the thermohaline circulation in the region (Zhang and Schmitt, 2000), affecting the water exchange between the Adriatic and Mediterranean.

The high vertical diffusivity coefficient of $5 \times 10^{-4}$ m$^2$/s found in Cardin et al. (2020) is explained by the dominant SF double-diffusion regime below 750 dbar, and the probability of SF occurring in the whole water column increased after the winter 2016/17.

All documented events occurred during a period of unprecedented high salinity transport from the Ionian to the Adriatic. The North Ionian Gyre (NIG) vorticity (not shown) decreased in the period from 2015-2018, and the system turned anticyclonic in

2017 (Civitarese et al., 2023, Mihanović et al., 2021). After 2018, the vorticity started to increase again and turned cyclonic in 2019. The eastward extension of the NIG between 2015 and 2019 was not enough to prevent the flow toward the Adriatic along the eastern board of the Ionian, conveying a large input of salt in the SAP (Fig. 2).

Measurements at the EMSO-E2M3A facility have demonstrated the Eulerian or fixed-point observatories importance as an essential component of the global ocean observing system. They allow detection of extreme events that occur rarely during the

multi-year observation period, such as the strong convection events that homogenize the water column leading to long term alterations of the density structure. They also offer several unique features not found in other systems and therefore complement



them (Cristini et al., 2016), and they provide a unique opportunity for multidisciplinary and interdisciplinary work combining a variety of observations on a large range of time scales.

**Author contribution**

All the authors contributed to the conceptualization, writing the original draft and review & editing. VC provided the funding acquisition.

**Competing interests**

The contact author has declared that none of the authors has any competing interest.

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

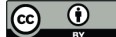



## Appendices




**Fig. A1: Time series of potential temperature (red) and salinity (blue) at the E2M3A site in the SAP.**



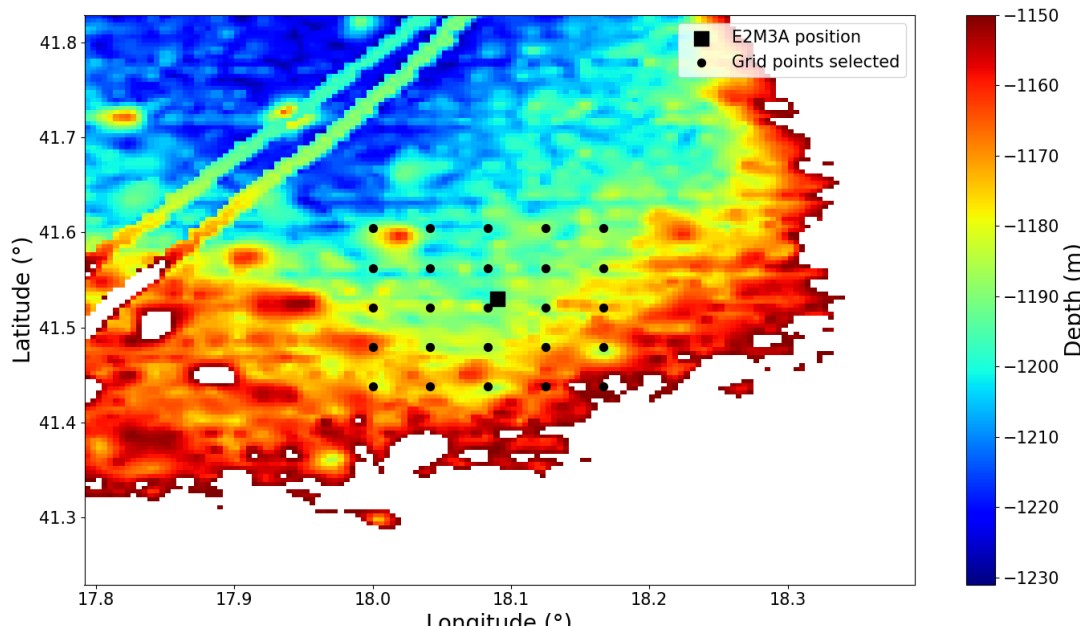

**Fig. A2: Selected grid points near the E2M3A position used to calculate the maximum depth of the mixed layer for the entire study period.**

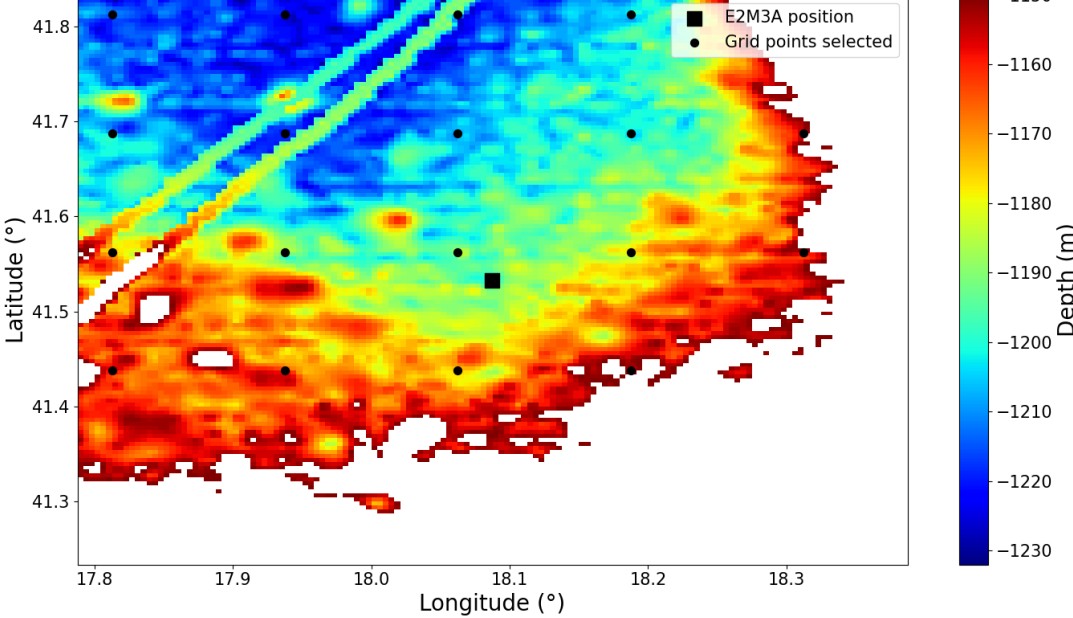

**Fig. A3 Selected grid points near the E2M3A position used to calculate the average relative vorticity for the entire study period.**



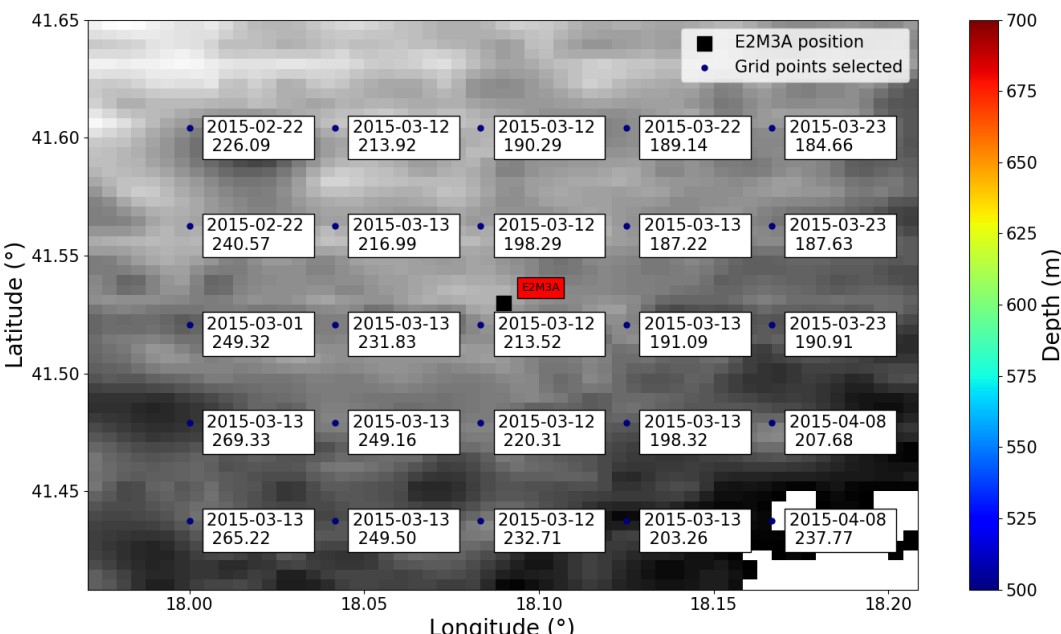

**Fig. A4: Selected grid points near E2M3A location used to determine the maximum depth of the mixed layer in winter 2015, as indicated in table 1.**

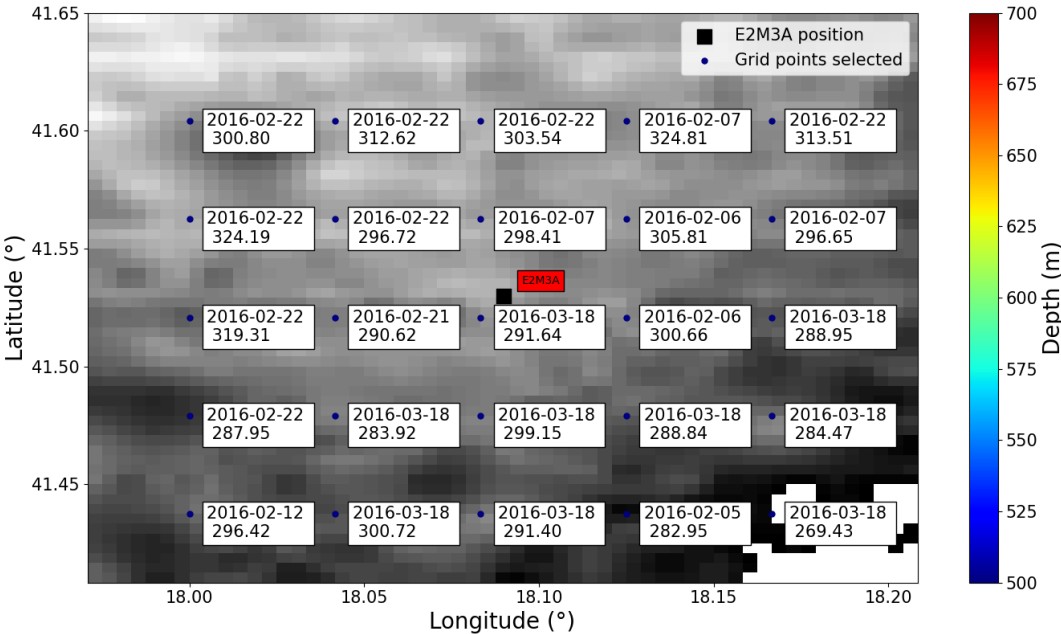

**Fig. A5: Selected grid points near E2M3A location used to determine the maximum depth of the mixed layer in winter 2016, as indicated in table 1.**




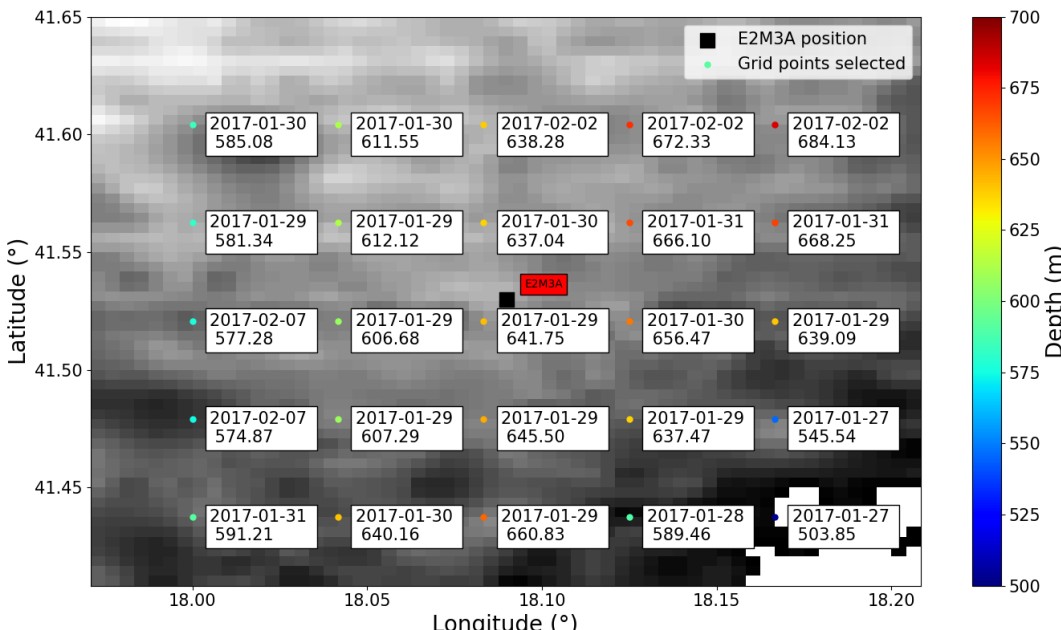

**Fig. A6: Selected grid points near E2M3A location used to determine the maximum depth of the mixed layer in winter 2017, as indicated in table 1.**

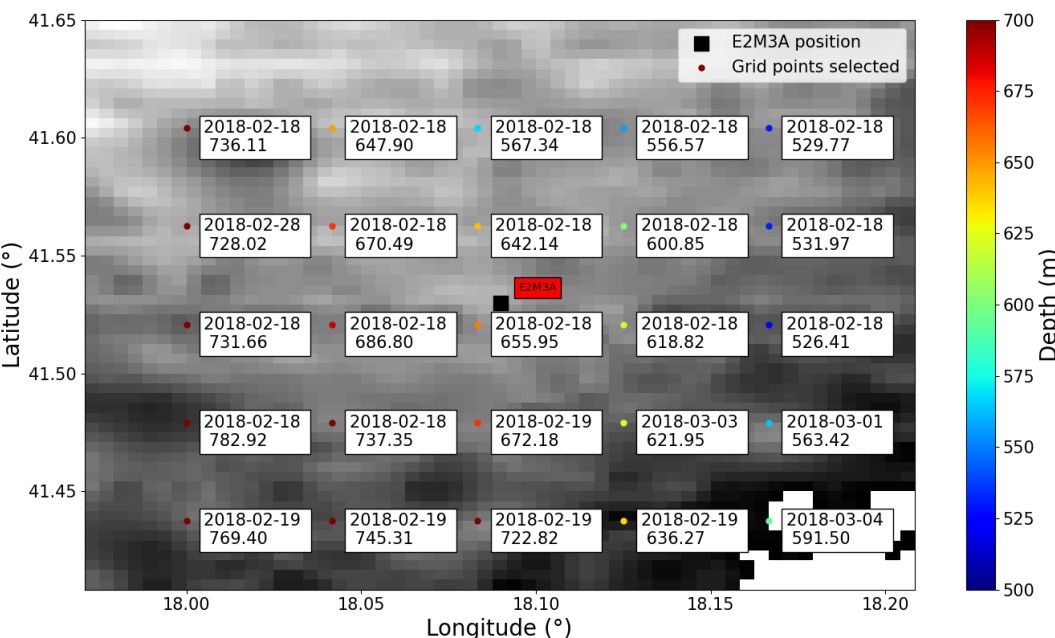

**Fig. A7: Selected grid points near E2M3A location used to determine the maximum depth of the mixed layer in winter 2018, as indicated in table 1.**



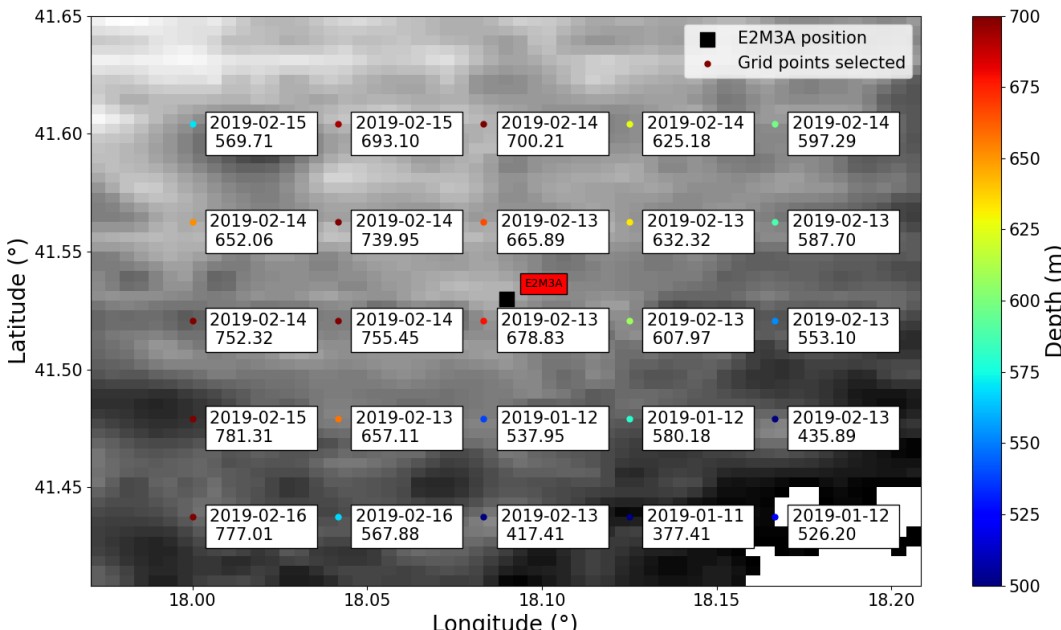

**Fig. A8: Selected grid points near E2M3A location used to determine the maximum depth of the mixed layer in winter 2019, as indicated in table 1.**

470