# Peer review of "Tipping of the double-diffusive regime in the Southern Adriatic pit in 2017 in connection with record high salinity values"

_EGUsphere, 2023_

## Referee Comment (RC2)

Title: Tipping of the double-diffusive regime in the Southern Adriatic pit in 2017 in connection with record high salinity values
Author(s): Felipe L. L. Amorim, Julien Le Meur, Achim Wirth, and Vanessa Cardin
MS No.: egusphere-2023-2481
MS type: Research article
Iteration: Initial submission
Special issue: Extremes in the marine environment: analysis of multi-temporal and multi-scale dynamics using observations, models, and machine learning techniques

Major comments :
The manuscript presents an original 5-year-long time series from the southern Adriatic that characterizes the evolution of the thermohaline content from the surface down to 1200 m. Despite the relative coarse vertical resolution of the measurements, the large scale variations of the salinity and temperature contents capture the winter deepening of the mixed layer, which depends on the intensity of surface heat losses and the salinity stratification. The analysis focuses on the potential role of the salt fingering regime on the evolution of the stratification of the region.

The pieces of information provided by this study are new, interesting for the region and complement previous studies. Therefore I would recommend the study to be published after some revisions.

My main comment is that, it's not because the stratification is prone to salt-fingering dynamics that salt fingers will indeed develop. Many times in the text, the opposite is mentioned, that is, authors observe a favorable regime for salt fingering, thus they conclude that there is mixing associated with that process. More care should be given in the use of the Turner angle.

At other places, the term "mixing" is used (good examples can be found in the start of the discussion). "Mixing" is a generic term that covers many processes. The process(es) should be mentioned whenever the term mixing is used so that the reader knows what it is all about (winter convection/deepening of the mixed layer, salt-fingering, …).

In the same vein, there are few descriptions of the figures done in the text that lack of precision and/or do not seem correct. Some examples are given in the detailed comments below.

Detailed comments:

l. 54: in the Med Sea there are several estimates of the diffusivities associated with "double-diffusion" that could be referenced on purpose (e.g. *Bryden, Harry L. et al. "Thermohaline staircases in the western Mediterranean Sea." Journal of Marine Research 72 (2014): 1-18*)

l. 120: How were the geostrophic velocities estimated at 1150 dbar ?

l. 146: I guess there is a connection between "the intrusion of high salinity into intermediate layers" and "the strong convection event", but the two are presented as independent observations…. Thus, I'm not sure that I have correctly understood if one is the consequence of the other, or if I do not focus on the right observations on Fig. 2…

l. 148: More care has to be given while referring to figures. As far as I can see on Fig. 3a, the heat loss do not exceed about 600 W m$^{-2}$ (and not 700) in January and 100 or 200 W m$^{-2}$ in March (and not 500). In Tab. 1, the max. heat loss of winter 2016-2017 is 623 W m$^{-2}$.

l. 178: Are you referring to isohalines (38.76 to 38.80) on Fig. 3b and Fig.2ab (38.78) ?

l. 181-182: "There is no convection during winter 2019…" ?!? If winter 2019 is winter 2018-2019, given Fig 2ab and Fig. 3ab , I would disagree.  The deepening of the mixed layer reaches 400-500 m. It is not as strong as the previous two winters but it is not very far behind.

l. 195-196: "...and the departure of the less saline… above the LIW (second)." This departure is not really visible on Fig. 3. I was wondering if referring to Fig. 2b would be better ? On Fig. 2b, we easily identify the salinity maximum associated with the LIW and the salinity minimum above in 2015 and 2016. After Winter 2016-2017, there is no more clear salinity maximum (LIW) and minimum (above).

l. 197: At a scale of a gyre or a meso-scale, cyclonic conditions can favor MLD deepening. Here there is a cyclonic vorticity observed. I was wondering what was the horizontal extent of the cyclonic circulation ? (and this also relates to my previous question on how was determined the vorticity, from which geostrophic current estimates ? On what scales ? Fig. A3 does not really helps in that matter).

l. 199: that "cyclonic preconditioning of the stratification + strong heat loss" favors a low N² due to deeper than usual convection, I agree. But I'm not sure that the same sequence implies "observed salt fingering". So far this is not observed and not described and this comment is confusing at this point of the manuscript. Later in section 3.3, you observe conditions that are somehow favorable to a salt fingering regime, though over most of the water column, the Turner angle is not strongly favorable (close to 90°). It remains weakly or moderately favorable, except in the 700-900 m layer where it is strongly favorable during year 2018-2019.

l. 254: "… the uppermost layer returns to strong SF": I guess this comment refers to the very short time peak in January (-February ?) 2017 that brings the turner angle close to 90° (since later on the turner angle is "only" weakly SF favorable). On the other hand, the vector length is very weak apart from a peak at the very beginning of January (from what I can guess looking at Fig. 5 and 6). If the vector length is weak, does the SF favorable peak in Turner angle matters ?

l. 226: "… leading to SF": How can you be sure that salt fingering is really actively occurring ? The sole observation of the Turner angle is insufficient.

l. 241: the paragraph starts with the two deepest layers? Then, second line we move to the top 3 layers (350-00). Third line, a prominent peak in Tu is mentioned, but is it that of layer 1 (350-550) occurring early 2017, or that of the lowest layer (1000-1200) in spring ? I'm a bit lost here.

l. 248: "… and a decrease in the two lowest layers": it is not clear for the deepest one. There is a increase in spring followed by some oscillations. I'm not sure about any decrease if I compare with years 2015-2016.

l. 249: wording… "The Tu shows significant destabilization….": What is significant ? What is "destabilization" ? Destabilization would mean instability. Again the Turner angle is not a proof that salt fingering is active. You could re-word in terms of "more or less favorable SF regime"..

l. 278: "we observe an increase of SF development"…. ??? Is there a figure showing that staircases increased ? You observe an increase of favorable conditions of the SF regime. You do not really observe the mixing associated with SF except for the example of staircases given with the Argo float, with the assumption that SF is at the origin of steps and layers.

l. 286: "reduced the mixing…" : Do you refer to winter convective mixing or salt fingering mixing here ?

l. 287: "… further reducing the vertical mixing": this would be right if SF favorable mixing conditions (as diagnosed with Tu angle) => active mixing. This is not the case. You can just say that it reduces the possibility of having active salt fingering.

l. 292 – 298: a schematic showing the different contributors acting on the stratification in the SAP would be nice to summarized these ideas. To avoid adding a supplementary Figure, I would suggest to replace Fig. 7 c-d-g-h-k-l by this schematics. Fig. 7 c-d-g-h-k-l (Turner angles) could be grouped with Fig. 7 a-b-e-f-i-j using a color scaling that depend on the Tu angle, or different markers depending on the Tu regimes.

l. 301: the vector length is a novelty, ok, but it is not that much used in this manuscript.

Minor comments:

- e.g. l. 51, 54, … end elsewhere: Some care to the formatting of units and numbers is needed (...x10-4 m2/s => … $\times 10^{-4}$ $m^2$ $s^{-1}$)(another example among others, caption of Fig. 3 "W*m-2"...)

l. 67: Could you add the seafloor depth at the mooring position ?

Fig. 3A, what are the red parts of the blue line ? This should be described in the caption.

l. 258: Fig. 7K instead of 8K ?

l. 289: suggestion "… due to the arrival of high saline surface waters that favored a convective…" (since its a part of the story, preconditioning by salinity, the other part being the enhanced heat loss compared with the two previous winters).

l. 409: Radko (not Ratko)

Figs A5-A8: what is the meaning of the color of the points ? (the only color-scale is depth).

---

## Author Comment (AC1)

Reply to Referee #2

Title: Tipping of the double-diffusive regime in the Southern Adriatic pit in 2017 in connection with record high salinity values

Author(s): Felipe L. L. Amorim, Julien Le Meur, Achim Wirth and Vanessa Cardin

MS No.: egusphere-2023-2481

MS type: Research article

Iteration: Initial submission

Special issue: Extremes in the marine environment: analysis of multi-temporal and multi-scale dynamics using observations, models, and machine learning techniques

**Major comments:**

**The manuscript presents an original 5-year-long time series from the southern Adriatic that characterizes the evolution of the thermohaline content from the surface down to 1200 m. Despite the relative coarse vertical resolution of the measurements, the large scale variations of the salinity and temperature contents capture the winter deepening of the mixed layer, which depends on the intensity of surface heat losses and the salinity stratification. The analysis focuses on the potential role of the salt fingering regime on the evolution of the stratification of the region.**

**The pieces of information provided by this study are new, interesting for the region and complement previous studies. Therefore I would recommend the study to be published after some revisions.**

We thank the Anonymous Referee #2 for the effort in reviewing the manuscript and for her/his positive evaluation. The posted comments and suggestions helped us to improve the manuscript.

**My main comment is that, it's not because the stratification is prone to salt-fingering dynamics that salt fingers will indeed develop. Many times in the text, the opposite is mentioned, that is, authors observe a favorable regime for salt fingering, thus they conclude that there is mixing associated with that process. More care should be given in the use of the Turner angle.**

The reviewer is right. Salt fingering (SF) Turner angle does not mean that there is necessarily SF happening (see also our answer to ref.3), it is a predisposition to SF. In scientific papers SF regime and SF Turner angle is often interchanged. Our mooring observation is spatially to coarse to directly observe SF stair cases. The assumption that there is a mixing process associated with salt fingers is based, as already mentioned, on the favorable Turner angle range (salt finger regime); salinity and temperature decreasing with depth; staircases in two vertical profiles of an Argo float and the high vertical diffusivity coefficient already studied by Cardin et al 2021, that could not be

explained at the time the article was published. We now added "a predisposition to SF" or "favorable to" in the manuscript at several locations:

**At other places, the term "mixing" is used (good examples can be found in the start of the discussion). "Mixing" is a generic term that covers many processes. The process(es) should be mentioned whenever the term mixing is used so that the reader knows what it is all about (winter convection/deepening of the mixed layer, salt-fingering, …).**

We agree and we updated the manuscript.

Line 7: "In double-diffusive mixing…"

Line 192: "…a convective mixing…"

Line 233: "…double-diffusion vertical mixing…"

Line 247: "…convective (winter convection) mixing."

Line 269: "…the favorable associated vertical mixing is weaker." Referring to SF.

Line 304: "…double-diffusive favorable mixing"

Line 306: "double-diffusion vertical mixing"

**In the same vein, there are few descriptions of the figures done in the text that lack of precision and/or do not seem correct. Some examples are given in the detailed comments below.**

**Detailed comments:**

**l. 54: in the Med Sea there are several estimates of the diffusivities associated with "double-diffusion" that could be referenced on purpose (e.g. Bryden, Harry L. et al. "Thermohaline staircases in the western Mediterranean Sea." Journal of Marine Research 72 (2014): 1-18)**

Thank you for the reference, it was added in the text.

Line 58: (Bryden et al. 2014; …)

**l. 120: How were the geostrophic velocities estimated at 1150 dbar ?**

There was a misunderstanding due to the form in which it was presented in the text. We meant that the surface relative vorticity (RV) was calculated in an area bounded by the 1150 isobath. The geostrophic velocities used to derive the RV are only on the surface and available from Copernicus Marine Service data.

Line 125: Surface relative vorticity were defined following Eq. (3):

$$\left(RV = \frac{\partial v}{\partial x} - \frac{\partial u}{dy}\right),$$

where u and v are surface geostrophic velocities (SEALEVEL_EUR_PHY_L4_MY_008_068 product; Copernicus Marine Service) in an area bounded by the isobath of 1150 dbar. The RV is the average of 18 grid points around the mooring position with 0.125° spatial resolution.

**l. 146: I guess there is a connection between "the intrusion of high salinity into intermediate layers" and "the strong convection event", but the two are presented as independent observations…. Thus, I'm not sure that I have correctly understood if one is the consequence of the other, or if I do not focus on the right observations on Fig. 2…**

The presence of higher salinity at surface followed by a strong convection event is seen as a combined process causing the extreme vertical change in salinity (see observation in the right at Fig 2). It should be noted that the two processes are not independent of each other, as the strong salinity signal at the surface interacts with the observed strong deep convection event. The results of the higher salinity at surface and the deep convection event are shown in Figure 3. We added:

Line 155: "…contributing to…"

**l. 148: More care has to be given while referring to figures. As far as I can see on Fig. 3a, the heat loss do not exceed about 600 W m-2 (and not 700) in January and 100 or 200 W m-2 in March (and not 500). In Tab. 1, the max. heat loss of winter 2016-2017 is 623 W m-2.**

We agreed with the reviewer and the sentence was altered.

Line 156: Indeed, strong heat losses occurred in January ($\approx$600 W/m$^2$; Table 1) (Fig. 3a), which, together with the contribution of salt in the water column, facilitated the erosion of the stratification (Fig. 3b, Table 1).

**l. 178: Are you referring to isohalines (38.76 to 38.80) on Fig. 3b and Fig.2ab (38.78) ?**

Yes, we included the figure reference in the text.

Line 180: (Fig. 3b)

**l. 181-182: "There is no convection during winter 2019…" ?!? If winter 2019 is winter 2018-2019, given Fig 2ab and Fig. 3ab , I would disagree. The deepening of the mixed layer reaches 400-500 m. It is not as strong as the previous two winters but it is not very far behind.**

The reviewer is right, convection took place even though not as strong as during the previous winters. The text was modified and rephrased in the manuscript:

Line 192: As a result, the amount of salt present in the water column triggered a convective mixing similar to that observed in the winter of 2017/18.

**l. 195-196: "...and the departure of the less saline… above the LIW (second)." This departure is not really visible on Fig. 3. I was wondering if referring to Fig. 2b would be better ? On Fig. 2b, we easily identify the salinity maximum associated with the LIW and the salinity minimum above in 2015 and 2016. After Winter 2016-2017, there is no more clear salinity maximum (LIW) and minimum (above).**

Agreed. We added the reference to Fig. 2.

Line 205: …with the high salinity below the surface (150 to 350 dbar) pushing the intrusion of the LIW core downward and the departure of the less saline water layers above the LIW (second) (Fig 2).

**l. 197: At a scale of a gyre or a meso-scale, cyclonic conditions can favor MLD deepening. Here there is a cyclonic vorticity observed. I was wondering what was the horizontal extent of the cyclonic circulation ? (and this also relates to my previous question on how was determined the vorticity, from which geostrophic current estimates ? On what scales ? Fig. A3 does not really helps in that matter).**

The horizontal scale of the cyclonic relative vorticity in the South Adriatic pit is about 100 km. The surface relative vorticity is the average of 18 grid points falling inside an area limited by the isobaths of 1150 dbar and derived from surface geostrophic velocities (daily and spatial resolution of 0.125°).

**l. 199: that "cyclonic preconditioning of the stratification + strong heat loss" favors a low $N^2$ due to deeper than usual convection, I agree. But I'm not sure that the same sequence implies "observed salt fingering". So far this is not observed and not described and this comment is confusing at this point of the manuscript. Later in section 3.3, you observe conditions that are somehow favorable to a salt fingering regime, though over most of the water column, the Turner angle is not strongly favourable (close to 90°). It remains weakly or moderately favorable, except in the 700-900 m layer where it is strongly favorable during year 2018-2019.**

We changed the sentence to:

Line 209: These three factors contributed to the observed low $N^2$ (Fig 4) and the SF favorable scenario.

**l. 224: "… the uppermost layer returns to strong SF": I guess this comment refers to the very short time peak in January (-February ?) 2017 that brings the turner angle close to 90° (since later on the turner angle is "only" weakly SF favorable). On the other hand, the vector length is very weak apart from a peak at the very beginning of January (from what I can guess looking at Fig. 5 and 6). If the vector length is weak, does the SF favorable peak in Turner angle matters ?**

Yes, we refer to the peak in January 2017. The peaks in Turner angle (Tu) and vector length (VL) match at this time. We observed after these peaks, when convection initiates, the sharp drop in TA, going to unstable regime, and reaching null values in VL. We rephrased the sentence.

Line 232: The stable stratification leads to low values of double-diffusion vertical mixing in the upper layer (350|550), and the underlying layers evolve more independently. Due to an increase in salinity at 350 dbar in early 2017, the uppermost layer showed peaks of strong SF and high VL. Subsequently, convection occurred causing the sharp decrease of VL and occurrence of unstable regime described by the Turner angle.

**l. 226: "… leading to SF": How can you be sure that salt fingering is really actively occurring ? The sole observation of the Turner angle is insufficient.**

As argued, we assume the vertical mixing by double-diffusion due to salinity and temperature gradients and the staircase features in the Argo floats profiles. Following the Reviewer suggestion, we modified the text.

Line 236: "…leading to favorable SF regime…".

**l. 241: the paragraph starts with the two deepest layers? Then, second line we move to the top 3 layers (350-00). Third line, a prominent peak in Tu is mentioned, but is it that of layer 1 (350-550) occurring early 2017, or that of the lowest layer (1000-1200) in spring ? I'm a bit lost here.**

Thank you for pointing this out. We refer to the first layer (350|550 dbar). We complemented the sentence.

Line 252: Analysis of VL shows an eradication of stratification in early 2017 from 350 dbar down to 900 dbar which persists until the end of the data record. The prominent peaks in Tu and VL in the first layer (350|550 dbar) in early 2017 are due to the arrival of warm, salty water at 350 dbar that possibly triggered strong SF (see also Fig. 7 and the discussion of the Argo data below).

**l. 248: "… and a decrease in the two lowest layers": it is not clear for the deepest one. There is a increase in spring followed by some oscillations. I'm not sure about any decrease if I compare with years 2015-2016.**

We agreed and we rephrased the sentence.

Line 260: The regime change in 2017 is indicated by an increase in Tu in the 750|900 dbar layer and a decrease in the 900|1000 dbar layer. In these two layers, the peaks of the pdfs are well separated, indicating a regime change. In the deepest layer (1000|1200 dbar), it is observed an oscillatory pattern from 2017, moving from SF to doubly stable regime, spreading the PDF in the second period but keeping the peaks close.

**l. 249: wording… "The Tu shows significant destabilization….": What is significant ? What is "destabilization" ? Destabilization would mean instability. Again the Turner angle is not a proof that salt fingering is active. You could re-word in terms of "more or less favorable SF regime"..**

Line 263: We rephrased to "The Tu shows notable tendency in SF in the 750|900 dbar layer after 2017 and a tendency to stabilization of the layer below, due to the convection event in 2017".

**l. 278: "we observe an increase of SF development"…. ??? Is there a figure showing that staircases increased ? You observe an increase of favorable conditions of the SF regime. You do not really observe the mixing associated with SF except for the example of staircases given with the Argo float, with the assumption that SF is at the origin of steps and layers.**

Line 296: "…we observed an increased predisposition of SF occurrence in the SAP due to the increase in salinity, and this could enhance salt-finger double diffusion in the subsurface layer of 350|550 dbar."

**l. 286: "reduced the mixing…" : Do you refer to winter convective mixing or salt fingering mixing here?**

We refer to double-diffusive mixing here. We added it to the text.

Line 304: From 2014 to the end of 2016 strong stratification above 400 dbar and at 800 dbar reduced the double-diffusive favorable mixing between the upper ocean and the deep SAP.

**l. 287: "… further reducing the vertical mixing": this would be right if SF favorable mixing conditions (as diagnosed with Tu angle) => active mixing. This is not the case. You can just say that it reduces the possibility of having active salt fingering.**

Line 304: After the winter of 2014/15, the upper layer passes from SF to doubly stably stratified, further reducing the possibility of having active salt fingering and consequently double-diffusion vertical mixing.

**l. 292 – 298: a schematic showing the different contributors acting on the stratification in the SAP would be nice to summarized these ideas. To avoid adding a supplementary Figure, I would suggest to replace Fig. 7 c-d-g-h-k-l by this schematics. Fig. 7 c-d-g-h-k-l (Turner angles) could be grouped with Fig. 7 a-b-e-f-i-j using a color scaling that depend on the Tu angle, or different markers depending on the Tu regimes.**

Thank you for the suggestion, we have discussed the point extensively and tried to come up with a scheme. However, the behavior differs every year (in winter), so we would need 5 different schemes. We would therefore very much prefer to keep the figures with the data and the histograms, which are essential to illustrate our point of tipping of the double diffusive regime as stated in the title. The text will then describe the processes acting at different times (years).

**l. 301: the vector length is a novelty, ok, but it is not that much used in this manuscript.**

VL is mentioned 22 times in the paper. It is important as it emphasizes the importance of Tu as discussed in the "Data and Methods" section. It is mentioned several times that stratification is low when VL is week, in this cases the Tu value is not significant. Analyzing Tu without considering VL is telling only part of the story.

**Minor comments:**

**- e.g. l. 51, 54, … end elsewhere: Some care to the formatting of units and numbers is needed (...x10-4 m2/s => … × 10-4 m2 s-1)(another example among others, caption of Fig. 3 "W*m-2"...)**

Thank you, we fixed the units along the text.

**l. 67: Could you add the seafloor depth at the mooring position ?**

The depth at the mooring position is around 1200 dbar, it was added to the text.

**Fig. 3A, what are the red parts of the blue line ? This should be described in the caption.**

Net heat flux. It was added to the caption.

**l. 258: Fig. 7K instead of 8K ?**

Yes, the reviewer is correct.

**l. 289: suggestion "… due to the arrival of high saline surface waters that favored a convective…" (since its a part of the story, preconditioning by salinity, the other part being the enhanced heat loss compared with the two previous winters).**

Thank you, we followed the suggestion.

**l. 409: Radko (not Ratko)**

Thank you, we changed in the reference part.

**Figs A5-A8: what is the meaning of the color of the points ? (the only color-scale is depth)**

The color of the points it the maximum MLD depth (blue to red colorbar) and the bathymetry is the background (white-black colorbar). Following suggestion from Referee #1, the images will be added to Supplementary material.

---

## Author Comment (AC2)

Reply to Referee #1

We thank the Anonymous Referee #1for the effort in reviewing the manuscript and for her/his positive evaluation. The posted comments have help us to improve the manuscript.

**The manuscript provides some calculations on the double diffusion or salt fingering processes in the Southern Adriatic Pit, which were somehow missing till now in the literature, although known to occur in the Adriatic and Mediterranean. The authors use long-term fixed mooring, in contrast to majority of similar studies which are based on the Argo profiling floats. I found the manuscript clearly written, focused and well organised, with just a few minor comments to raise:**

Thank you very much for the positive evaluation. We made changes following the Referee comments and suggestions.

**- Fig. 1. The figure is hard to read, for example positions of Argo profiling float. It might be better to have 2-D plot with clear colours, e.g. blue for bathymetry, dashed arrows for deep currents, etc.**

Thanks to the reviewer for the suggestion. A new figure with more information in 2-D has been added substituting the previous one.

**- Lines 106-111. The authors claim that introduction of VL variable in this study is something innovative. However, I cannot get this from the manuscript - maybe to explain or justify this (Why is innovative? Maybe a bit better explain the variable), here or in results where you present the variable.**

The VL is a key quantity, it is used 22 times in the paper. When introduced in the paper (Data and Methods section), it was and is written:

Line 111: "To evaluate further the water mass-properties in the vertical, we calculated the vector length (VL) defined following Eq. (2):

$$VL = \sqrt{\left(\alpha \frac{d\theta}{dz}\right)^2 + \left(\beta \frac{dS}{dz}\right)^2}.$$

To the best of our knowledge, the VL has not been discussed in connection with stratification and the Tu. A higher VL indicates an increased change of water-mass properties and therefore emphasizes the importance of Tu. On the other hand, when the VL is small the water column is essentially unstratified and changes in the Tu insignificant."

We now added:

Line 116: A water mass is characterized by (T,S), its temperature and salinity. The variables (Tu,VL) are simply the polar coordinates in (T,S) space. It is Tu that determines the stability regime and VL the significance.

**- Figs A2 to A5 might be better to place as supplementary material, not to weighten the article as appendix, Fig. A1 looks nice and would be better to be placed in the main article.**

Figs A2 to A8 (now A1 to A7) will be placed as supplementary material, as suggested by the reviewer. Fig A1 has been merged with Fig 2.

---

## Author Comment (AC3)

Reply to Referee #3

**The manuscript analyses a temperature and salinity dataset, obtained from a mooring in the southern Adriatic, at seven depth levels from surface to bottom, from November 2014 to October 2019. With the help of additional data from ECMWF and Copernicus and the profiles of an Argo float, the authors describe the evolution of thermohaline properties in the water column, relating them to heat fluxes at the air-sea surface, and analyzing them towards other variables, such as mixed layer depth, relative vorticity and Turner angle.**

**Unfortunately, the study contains improper generalizations and some inaccuracies, already highlighted in particular by reviewer 2. Among these, the most serious is the recurring confusion between "predisposition to" and "occurrence of" salt fingering process, and a somewhat forced use of the Turner angle, compared to its original definition (Ruddick, 1983). Many arguments are based on this confusion and are therefore far from being proven valid.**

Referee #3 (Ref. 3) is right when she/he says that with our vertically coarse mooring data only allows us to determine the "predisposition to" and not the "occurrence of" salt-fingering. However, to the best of our knowledge there is no long-term (multi-year), fine-resolution spatial and temporal data that could provide a definitive answer. Ref. 3 also refers to the work of Ruddick (1983), but Ruddick (1983) cites regions that are "salt fingering" when Turner angle (Tu) indicates this. Furthermore, Ruddick (1983) begins in line 2 with "The single most important external parameter that indicates the relative strength of double-diffusion is the gradient ratio ..." (the Tu is an amelioration of the gradient ratio and directly linked to it). In the rest of his work, especially in the last paragraph, Ruddick (1983) does not distinguish between "predisposition to" and "occurrence of". In our work, we are more cautious, corroborating our results with a high mixing coefficient, found in Cardin et al. (2020) and with staircase observations in the ARGO float data. We also complement Tu with vector length (VL), suggesting that salt fingering at high VL values are more prone to effectively produce salt fingering.
To acknowledge the comments of Ref. 3, we have now changed the assumption of occurrence to "favorable", "predisposition" or "possible" conditions for salt fingering occurrence.

**In my opinion, for possible publication, the work requires further analysis or a new goal setting, considering that double diffusion processes, and SF in particular, are the core theme of the present study.**

Thank you for the feedback. We now consider that the Turner angle will point favorable or predisposing conditions of determined double-diffusion regime. We remain confident that the goal mentioned in the title: analyzing the SF and showing a tipping in 2017, is well argued in our paper.

**Below are some comments in addition to what the other two reviewers have already noted.**

**Line 7**. **The opening sentence of the abstract contains a dubious generalization. Perhaps salinity was confused with density?**

Thank you for pointing this. We are referring to double-diffusion processes that occur due to differences in the diffusivity coefficients, not general instability caused by density gradients. We rephrased the sentence.

Line 7: "In double-diffusive mixing, whenever salinity and temperature decrease with depth, the water column is either unstable or favorable to a state called salt fingering (SF)…"

**Line 41**. **Durante et al. (2019) did not use Argo data but CTD profiles from cruises. A more appropriate reference for this topic is Taillandier et al. (Biogeosciences 17, 3343–3366. doi: 10.5194/bg-17-3343-2020) who used CTD and ARGO profiles, from the Tyrrhenian Sea and the Algerian Basin.**

Thank you for pointing this mistake. We fixed the text and included the new reference.

Line 43: Durante et al. (2019) used a longer time series of CTD casts and Menna et al. (2021) analyzed years of Argo data profiles in the Tyrrhenian and Ionian/Levantine Seas, respectively, with the temporal resolution of the analyzed data ranging from weeks to months. Taillandier et al. (2020) used a combination of about 700 CTD and Argo floats profiles collected from 2013 to 2017 to study thermohaline staircases related to double-diffusion in the Western Mediterranean sea.

**Line 93. "In order to explore if the water column was undergoing double-diffusive convection and its related local stability we estimated the Turner angle". Wrong approach: with Tu you only evaluate whether the water column is inclined to a given regime and do not demonstrate its actual presence.**

The Turner angle is the principal indicator to determine if a predisposition to SF exists, so our approach is correct. We agree that the wording was not precise and now changed the sentence to:

Line 98: In order to explore if the water column was favorable to double-diffusive regimes and its related local stability possible condition we estimated the Turner angle (Tu) defined following Eq.(1):

**Line 95. Equation 2: missing brackets make the formula incorrect**

$$Tu = tan^{-1}\left(\frac{\alpha\partial\theta}{\partial z} - \frac{\beta\partial S}{\partial z}, \frac{\alpha\partial\theta}{\partial z} + \frac{\beta\partial S}{\partial z}\right);$$

Thank you. Fixed.

**Line 106. A reference is missing for the vector length (VL). If it is your original introduction of a new analysis parameter you must explain it better.**

Thank you, we consider VL a new analysis parameter. It was and is written in the paper:

Line 114: "To the best of our knowledge the VL has not been discussed in connection with stratification and the Tu."

we now added:

Line 116: A water mass is characterized by ($\theta$, S), its potential temperature and salinity. The variables (Tu, VL) are simply the polar coordinates in ($\theta$, S) space. It is Tu that determines the stability regime and VL the significance.

**Lines 118-120. Can you explain better how you calculate geostrophic velocities and RV? Which dataset do you use? Perhaps something should be guessed from the figures A2-A7, but these also require some explanation.**

The surface geostrophic velocities were obtained from the Copernicus Marine Service, SEALEVEL_EUR_PHY_L4_MY_008_068 product. We added to the text.

Line 125: Surface relative vorticity was defined following Eq. (3):

$$(RV = \frac{\partial v}{\partial x} - \frac{\partial u}{\partial y}),$$

where u and v are surface geostrophic velocities (SEALEVEL_EUR_PHY_L4_MY_008_068 product; Copernicus Marine Service) in an area limited by the isobath of 1150 dbar. The RV is the average of 18 grid points around the mooring position with 0.125° spatial resolution.

**Line 263. Instead of Durante et al. (2019), I would cite Durante et al. (2021, Front. Mar. Sci. 8:672437. doi: 10.3389/fmars.2021.672437) which is more appropriate in the context of fluxes.**

Thank you and we accepted the suggestion.

---

## Referee Report (RR1)

Title: Tipping of the double-diffusive regime in the Southern Adriatic pit in 2017 in connection with record high salinity values
Author(s): Felipe L. L. Amorim, Julien Le Meur, Achim Wirth, and Vanessa Cardin
MS No.: egusphere-2023-2481
MS type: Research article
Iteration: Revised submission
Special issue: Extremes in the marine environment: analysis of multi-temporal and multi-scale dynamics using observations, models, and machine learning techniques

Major comments :
The manuscript has been corrected according to the comments received for the initial version. Especially, more care is given to the question of salt-fingering (SF) probability of occurrence. However, I still have a few comments as you will see below.

Abstract:
The abstract concludes with "Consequently, we observe an alteration of vertical stratification throughout the water column". This adverb, "consequently", arises just after two sentences describing the increased predisposition to SF. Thus, the reader gets the impression that the alteration of vertical stratification **throughout** the water column may be due to SF while it is clearly due to convection after winter heat loss at the surface. This sounds misleading.

l. 23: parentheses should be removed.

l. 55: correct me if I am wrong but the bulk diffusivity coefficient of 5x10-4 m2 s-1 based on two types of data from Cardin at al. (2020) encompasses all dynamical processes acting for years and not only double diffusion activity (for example inertial waves, courant-topography induced mixing, ...). Because of previous sentences and the way it is stated, the reader gets the idea that double diffusion is responsible for this eddy value. More care is needed in the formulation if I am right, and more details should be given when referring to Cardin at al. (2020). Moreover, Cardin at al. (2020) focused on regions deeper than 750 dbar while this manuscript often focuses on the regions above.

l. 58: it is true that the "bulk" (i.e. calculated over large vertical scales of several hundreds of meters) eddy diffusivity coefficient of tracers (temperature, salinity) are around 2-6 x10-4 m2 s-1. Those values come out when the large-scale tracer gradient are taken for the computation. However, this large-scale gradient encompasses steps/interfaces (strongly stratified with double diffusive activity) and layers (almost unstratified). As Bryden et al. (2014) stated, double diffusive processes operate on the thin steps (meter-scale). There, $kS = 3.7 \times 10^{-5}$ m2 s−1 and $kT = 2.0 \times 10^{-5}$ m2 s−1, that is 15 times weaker than the bulk estimates. Those eddy coefficients are really those associated with the double-diffusive activity and were also measured with microstructure data (Schmitt et al., 2005, doi:10.1126/science.1108678; Ferron et al. 2021, doi: 10.3389/fmars.2021.664509) and in Radko and Smith (2012, doi:10.1017/jfm.2011.343)'s model for instance. Those eddy diffusivities remain small, but are associated with interfaces ("strong" vertical property gradients, which is the other important parameter when computing turbulent property fluxes) and, more importantly, concerns most of the time large spatial areas (as assumed at least in the Med, but also in some regions of the Atlantic for example). Thus, the overall contribution of double diffusive processes is thought to have an impact at

the scale of those large areas, especially when they act for a long time (as assumed for the Tyrrhenian Sea). Maybe that a bit more detail and caution is needed when giving the eddy diffusivity values.

l. 309: 'that' missing ? '… due to the arrival of high saline waters **that** favored a convective event penetrating'

l. 313: 'possible' missing: ' Salt fingering is shown to be a consequence and a **possible** driver of the density structure…'

l. 317: 'How the competition between sporadic-extreme and continuous processes changes the structure of the water column and, in the future, how it acts on the thermohaline circulation of the Mediterranean is not decided.'
→ to gain insight on this competition for the SAP, you could apply a rapid calculation of salt and heat fluxes using bulk eddy coefficient typically encountered when SF is supposed to be active, and compare them to those due to the fast convection events. Given various assumptions regarding the duration and the vertical extension of assumed SF activity, you would get a rough idea of the competition between sporadic-extreme convection and more or less 'continuous' processes associated with SF processes.

l. 325: 'The high vertical diffusivity coefficient of 5x10-4 m2 s-1 found in Cardin et al. (2020) is explained by the dominant SF double-diffusion regime below 750 dbar, and the probability of SF occurring in the whole water column increased after the winter of 2016/17.'

Several points:
1- There is an ambiguity:
When I read this, I wonder whether the explanation comes from Cardin et al. (2020) or from your study and conclusions. From my (too) fast reading of Cardin et al. (2020), I would say this is your assumption.

2- Again, if my reading of Cardin et al. (2020) was not too rapid, their reported eddy diffusivity only concerns waters deeper than 750 m. You should then be more specific. Given Fig. 5, it appears safe enough to assume that double diffusion (SF) may be the main contributor of this diffusion in the depth range (750-900). However, above 750 m, Turner angles are rarely very favorable to SF and your assumption is then quite weak and it sounds more like a speculation.

3- on the opposite, to support your SF assumption, you could use the two Argo profiles and calculate the Turner angle across the interfaces in regions where staircases were observed. Indeed, in the first version of your manuscript, the Turner were only moderately favorable (50-70°) to salt-fingering, but not that close to very favorable (close to 90°). I was wondering if there was any sensitivity in the way you calculated those Turner angles with the Argo profiles (which were smoothed over 20 m or so instead of calculating them across the identified interfaces in the staircase structures?). If, you could obtain Turner angles closer to 90°, this would bring some support to SF activity and contribution, at least in the regions where staircases were observed. And then you have an argument to say that the very limited vertical resolution of the mooring, that shows weak to moderate favorable angles to salt-fingering in the upper layers, may hide some much more favorable angles to SF (if you had more resolution to compute them).

---

## Author Response (AR2)

Dear Editor,
below you will find our responses to the comments from reviewer 2. We have also added a further reference to the use of ECMWF data and corrected the multiple references chronologically.
Yours sincerely,

Vanessa Cardin

Title: Tipping of the double-diffusive regime in the Southern Adriatic pit in 2017 in connection with record high salinity values
Author(s): Felipe L. L. Amorim, Julien Le Meur, Achim Wirth, and Vanessa Cardin
MS No.: egusphere-2023-2481
MS type: Research article Iteration:
Revised submission
Special issue: Extremes in the marine environment: analysis of multi-temporal and multi-scale dynamics using observations, models, and machine learning techniques

We thank the reviewer 2 again for his corrections. Please find our answers in blue and the corrections performed in **red**. A revised version with our corrections in **color** is also provided.

Major comments :
The manuscript has been corrected according to the comments received for the initial version. Especially, more care is given to the question of salt-fingering (SF) probability of occurrence. However, I still have a few comments as you will see below.

Abstract:
The abstract concludes with "Consequently, we observe an alteration of vertical stratification throughout the water column". This adverb, "consequently", arises just after two sentences describing the increased predisposition to SF. Thus, the reader gets the impression that the alteration of vertical stratification **throughout** the water column may be due to SF while it is clearly due to convection after winter heat loss at the surface. This sounds misleading.

We removed:

l. 23: parentheses should be removed.

Thanks for pointing this out, the brackets have been removed.

l. 55: correct me if I am wrong but the bulk diffusivity coefficient of 5x10-4 m2 s-1 based on two types of data from Cardin at al. (2020) encompasses all dynamical processes acting for years and not only double diffusion activity (for example inertial waves, courant-topography induced mixing, ...). Because of previous sentences and the way it is stated, the reader gets the idea that double diffusion is responsible for this eddy value. More care is needed in the formulation if I am right, and more details should be given when referring to Cardin at al. (2020). Moreover, Cardin at al. (2020) focused on regions deeper than 750 dbar while this manuscript often focuses on the regions above.

We now added:
**in the deep SAP (below the sill depth of 750 m)**

It was and is written in the paper that:
"can enhance vertical mixing through double diffusion"

and further down (l. 59)
"it can affect larger spatial extents by mixing water mass properties"

in l. 62:
"can enhance the mixing"

in l. 63:
"their potentially important contribution to vertical mixing"

We do not feel that due to our formulation "the reader gets the idea that double diffusion is the only responsible for this eddy value"

l. 58: it is true that the "bulk" (i.e. calculated over large vertical scales of several hundreds of meters) eddy diffusivity coefficient of tracers (temperature, salinity) are around 2-6 x10-4 m2 s-1. Those values come out when the large-scale tracer gradient are taken for the computation. However, this large-scale gradient encompasses steps/interfaces (strongly stratified with double diffusive activity) and layers (almost unstratified). As Bryden et al. (2014) stated, double diffusive processes operate on the thin steps (meter-scale). There, $kS = 3.7 \times 10^{-5}$ m2 s$-1$ and $kT = 2.0 \times 10^{-5}$ m2 s$-1$, that is 15 times weaker than the bulk estimates. Those eddy coefficients are really those associated with the double- diffusive activity and were also measured with microstructure data (Schmitt et al., 2005, doi:10.1126/science.1108678; Ferron et al. 2021, doi: 10.3389/fmars.2021.664509) and in Radko and Smith (2012, doi:10.1017/jfm.2011.343)'s model for instance. Those eddy diffusivities remain small, but are associated with interfaces ("strong" vertical property gradients, which is the other important parameter when computing turbulent property fluxes) and, more importantly, concerns most of the time large spatial areas (as assumed at least in the Med, but also in some regions of the Atlantic for example). Thus, the overall contribution of double diffusive processes is thought to have an impact at the scale of those large areas, especially when they act for a long time (as assumed for the Tyrrhenian Sea). Maybe that a bit more detail and caution is needed when giving the eddy diffusivity values.

The reviewer is right, the problem of how gradients and turbulent motion and different scales interact to cause mixing, and how this can be cast into a simple diffusion equation with a scale dependent diffusivity, is the coarse-graining problem. The coarse-graining problem has so far only been solved for very simplified extended systems (spin classes) using renormalization group methods. These methods are gradually being applied to diffusion problems in homogeneous isotropic turbulence, but are still far from providing insights into real data. In the SAP, the situation is inhomogeneous, non-isotropic and a large number of processes interact non-linearly. We do not want to mention this topic in this paper, as it only causes confusion. In Cardin et al. (2020) and in the present paper, there is no confusion because the estimates are based on data separated by O(100 m) in the vertical. Thus, we only can and only speak of large-scale ( bulk ) coefficients. We have now added:

**large-scale (bulk)**

l. 309: 'that' missing ? '… due to the arrival of high saline waters **that** favored a convective event penetrating'

We added **that**.

l. 313: 'possible' missing: ' Salt fingering is shown to be a consequence and a **possible** driver of the density structure…'

We added **possible**.

l. 317: 'How the competition between sporadic-extreme and continuous processes changes the structure of the water column and, in the future, how it acts on the thermohaline circulation of the Mediterranean is not decided.'
→ to gain insight on this competition for the SAP, you could apply a rapid calculation of salt and heat fluxes using bulk eddy coefficient typically encountered when SF is supposed to be active, and compare them to those due to the fast convection events. Given various assumptions regarding the duration and the vertical extension of assumed SF activity, you would get a rough idea of the competition between sporadic-extreme convection and more or less 'continuous' processes associated with SF processes.

This was done in Cardin et al. (2020), where we constructed a 1D forced diffusion model and estimated the diffusion parameter, which was still unexplained at that time. We also determined a characteristic time scale for the diffusion process in the dSAP of about 7 years. In the present work, we evaluate SF as an important factor. The forcing due to gravity currents is the subject of current research in the ongoing PhD thesis of J. Le Meur (co-author). However, the aim of this work is to use mainly the E2M3A high-frequency time series.

The text has been changed to:

How the competition between sporadic-extreme and continuous processes changes the structure of the water column and how it acts on the thermohaline circulation of the Mediterranean will be a subject of future studies.

l. 325: 'The high vertical diffusivity coefficient of 5x10-4 m2 s-1 found in Cardin et al. (2020) is explained by the dominant SF double-diffusion regime below 750 dbar, and the probability of SF occurring in the whole water column increased after the winter of 2016/17.'

Several points:
1-      There is an ambiguity:
When I read this, I wonder whether the explanation comes from Cardin et al. (2020) or from your study and conclusions. From my (too) fast reading of Cardin et al. (2020), I would say this is your assumption.

One of the main goals of this work is to explain the high vertical diffusivity coefficient found in Cardin et al. (2020). We believe that this has been achieved with our analysis, in which it is mainly explained by the dominant SF double diffusion regime below 750 dbar, and that the probability of SF occurrence in the entire water column has increased after the winter of 2016/17.

The text has been changed to:

The high vertical diffusivity coefficient of 5x10-4 m2 s-1 found in Cardin et al. (2020) is here explained by the dominant SF double-diffusion regime below 750 dbar, and the probability of SF occurring in the whole water column increased after the winter of 2016/17.

2-      Again, if my reading of Cardin et al. (2020) was not too rapid, their reported eddy diffusivity only concerns waters deeper than 750 m. You should then be more specific. Given Fig. 5, it appears safe enough to assume that double diffusion (SF) may be the main contributor of this diffusion in the depth range (750-900). However, above 750 m, Turner angles are rarely very favorable to SF and your

assumption is then quite weak and it sounds more like a speculation.

Indeed, Cardin et al. (2020) base their analysis on two types of data, namely 13-year time series of observational data (2006–2019) of temperature from the E2M3A Observatory and 55 vertical profiles (1985–2019) below 750 m depth in the dSAP. The aim of extending this work to the layers above this depth was to observe the effects of deep convection and salt finger conditions. The citation of the reviewer includes "below 750 dbar" so we explicitly state that it is about the deep SAP.

3- on the opposite, to support your SF assumption, you could use the two Argo profiles and calculate the Turner angle across the interfaces in regions where staircases were observed. Indeed, in the first version of your manuscript, the Turner were only moderately favorable (50-70°) to salt-fingering, but not that close to very favorable (close to 90°). I was wondering if there was any sensitivity in the way you calculated those Turner angles with the Argo profiles (which were smoothed over 20 m or so instead of calculating them across the identified interfaces in the staircase structures?). If, you could obtain Turner angles closer to 90°, this would bring some support to SF activity and contribution, at least in the regions where staircases were observed. And then you have an argument to say that the very limited vertical resolution of the mooring, that shows weak to moderate favorable angles to salt- fingering in the upper layers, may hide some much more favorable angles to SF (if you had more resolution to compute them).

The strongest evidence for SF is staircases. Therefore, the Argo data was used to show that staircases occur where they are predicted by our coarse-resolution TU analysis. Thus, we make clear that vertically coarse resolution data can be used to indicate the possible occurrence of SF. A detailed investigation of TU variations across staircases is not possible with our mooring observations and is not the subject of this paper (although it is undoubtedly an interesting topic).